# An ELOVL2-Based Epigenetic Clock for Forensic Age Prediction: A Systematic Review

**DOI:** 10.3390/ijms24032254

**Published:** 2023-01-23

**Authors:** Ersilia Paparazzo, Vincenzo Lagani, Silvana Geracitano, Luigi Citrigno, Mirella Aurora Aceto, Antonio Malvaso, Francesco Bruno, Giuseppe Passarino, Alberto Montesanto

**Affiliations:** 1Department of Biology, Ecology and Earth Sciences, University of Calabria, 87036 Rende, Italy; 2Biological and Environmental Sciences and Engineering Division (BESE), King Abdullah University of Science and Technology (KAUST), Thuwal 23952, Saudi Arabia; 3Institute of Chemical Biology, Ilia State University, Tbilisi 0162, Georgia; 4SDAIA-KAUST Center of Excellence in Data Science and Artificial Intelligence, Thuwal 23952, Saudi Arabia; 5National Research Council (CNR)-Institute for Biomedical Research and Innovation–(IRIB), 87050 Mangone, Italy; 6Department of Brain and Behavioral Sciences, IRCCS “C. Mondino” Foundation, National Neurological Institute, University of Pavia, 27100 Pavia, Italy; 7Regional Neurogenetic Centre (CRN), Department of Primary Care, ASP Catanzaro, 88046 Lamezia Terme, Italy; 8Association for Neurogenetic Research (ARN), 88046 Lamezia Terme, Italy

**Keywords:** ELOVL2, epigenetic clock, age prediction, methylation, pyrosequencing

## Abstract

The prediction of chronological age from methylation-based biomarkers represents one of the most promising applications in the field of forensic sciences. Age-prediction models developed so far are not easily applicable for forensic caseworkers. Among the several attempts to pursue this objective, the formulation of single-locus models might represent a good strategy. The present work aimed to develop an accurate single-locus model for age prediction exploiting *ELOVL2*, a gene for which epigenetic alterations are most highly correlated with age. We carried out a systematic review of different published pyrosequencing datasets in which methylation of the *ELOVL2* promoter was analysed to formulate age prediction models. Nine of these, with available datasets involving 2298 participants, were selected. We found that irrespective of which model was adopted, a very strong relationship between *ELOVL2* methylation levels and age exists. In particular, the model giving the best age-prediction accuracy was the gradient boosting regressor with a prediction error of about 5.5 years. The findings reported here strongly support the use of *ELOVL2* for the formulation of a single-locus epigenetic model, but the inclusion of additional, non-redundant markers is a fundamental requirement to apply a molecular model to forensic applications with more robust results.

## 1. Introduction

The prediction of chronological age from methylation-based biomarkers represents one of the most promising potential applications in the field of forensic sciences [1,2,3]. In the last decade, extensive efforts have been made to identify such biomarkers and thanks to several epigenome-wide association studies, many CpG sites for which methylation levels are strongly correlated with age have been identified. Several authors proposed to combine these markers to formulate models for age prediction [4,5,6]. Currently, the most robust methylation-based age prediction methods are represented by the so-called “epigenetic clocks” that are based on microarray technologies. These methods are technically not achievable in a typical forensic laboratory and require more DNA than the usual amount available for most casework samples. In addition, they are based on technologies that are very expansive and complex also provide very sophisticated classification algorithms. More recently, forensic DNA technology has triggered efforts toward simplification of the array-based epigenetic clocks, and several models have been developed to date. Due to the existence of complex nonlinear relationships between the methylation levels of the assessed CpG markers and chronological age, several authors have also taken advantage of machine learning approaches to obtain more accurate age predictions [7,8]. These algorithms include support vector machine (SVM) [9], artificial neural networks [10], gradient boosting regressor [11], and missMDA [8]. However, their translation to forensic genetic practices is still far away for several important reasons. Firstly, with some exceptions, they are still based on technological formats that are not available in a typical forensic genetic laboratory. Secondly, even if formulated to exploit the same technology, the detection of the methylation values might be based on different genes, and therefore, the CpG sites (and their combination) included in such models are usually different. Thirdly, the complexity of the statistical framework within which these models have been obtained limits their application.

The lack of a consensus on both a standard set of CpG markers to use for practical applications and a feasible and accurate tool for assessing the methylation levels based on this standard set of markers make it difficult to compare the results across different studies. The development of single-locus age-prediction models might represent one good strategy to cope with these kinds of problems. In this context, ELOVL fatty acid elongase 2 (ELOVL2) represents a robust candidate gene as (i) its epigenetic variability is highly correlated with age predictions [12,13,14,15,16,17], (ii) it is included in most current age prediction models for forensic applications [8], and (iii) it does not show tissue-specificity, as observed for most of the epigenetic markers identified so far [18]. For these characteristics, a single-locus model might also allow for the development of a simpler prediction model that based on an easy technology format, could be easily implemented in forensic laboratories.

The present study aimed to develop an easy, robust, and improved blood-based age prediction model using ELOVL2 promoter methylation data. To this purpose, we carried out a systematic review of different previously published pyrosequencing datasets of ELOVL2 promoter DNA methylation assessing the relationship between ELOVL2 methylation levels and the age of the recruited subjects.

## 2. Materials and Methods

This systematic review was conducted following the Preferred Reporting Items for Systematic Review and Meta-Analysis (PRISMA) guidelines [19]. We registered the protocol in the International Platform of Registered Systematic Review and Meta-analysis Protocols (registration number: INPLASY2022120006).

### 2.1. Study Search

A systematic search was carried out using SCOPUS and PUBMED databases. A manual search of the bibliographies of selected articles was also conducted. The following Boolean search strings were used considering free text and the Medical Subject Heading (MeSH) (Table 1):

All returned results were systematically identified, screened, then extracted for relevant information following the PRISMA guidelines [19].

### 2.2. Exclusion and Inclusion Criteria

All studies with the aim of understanding the relationship between the *ELOVL2* methylation levels and age written in English language, carried out on humans and providing a publicly available dataset, were included in the systematic review. Articles that did not include original research (e.g., review, opinion article, or conference abstract) and for which methylation analysis was carried out using a technology different from pyrosequencing or in tissues different form blood were excluded from further analyses. Titles, abstracts, and articles were evaluated by two separate reviewers (E.P. and A.A.). Titles and abstracts were reviewed for subject relevance. The investigators read full-text versions of eligible articles on their own. Disagreements were addressed based on a consensus between the two reviewers. A third investigator (A.Mo.) was consulted if the two reviewers reached different decisions or when in doubt.

### 2.3. Statistical Analysis

For modelling based on the *ELOVL2* methylation status, nine publicly available datasets (see Table 2) were merged. The resulting comprehensive dataset (2298 subjects in total) was randomly split in half, creating a training and validation set. The split was performed in such a way to obtain the same age distribution between the training and test set. The training set was then used for creating several predictive models, described below, which were able to provide age estimates for single individuals, while the validation set was used to unbiasedly assess the predictive capabilities of each model.

Sensitivity analysis was performed using a leave-dataset-out approach [26]. This means repeating the above procedure nine times, each time holding one dataset out of both the training and validation set. The hold-out dataset was then used for further assessing the performance of the best model found based on the training/validation set.

We tested five different statistical approaches for the development of *ELOVL2* single-locus age prediction models. The first one consisted of a multiple linear regression (MLR) model using the single loci as independent variables, as proposed by Zbieć-Piekarska and colleagues [15]. The second approach was a multiple quadratic regression (MQR), including the single loci along with their pairwise interactions and their square powers. This model was chosen because the methylation levels of some CpGs from *ELOVL2* might be better modelled using a quadratic model, thereby improving the age-prediction accuracy of the model [21]. The third approach was the SVM for regression with a Gaussian kernel, a non-linear machine learning algorithm able to capture complex patterns and interactions within data [27]. Gradient boosting regression (GBR) combines a series of regression models together, and each model trained to improve upon the error of its predecessor. Both SVM and MQR were the best performing models in the study carried out by Garali and co-workers [16]. Finally, we trained a multiple linear model using the first principal component (PC) derived through principal components analysis (PCA). The model included the PC, as well as its exponentiations up to the 4th degree. For ensuring an unbiased assessment of this approach, the PC loadings derived from the training set were used for computing the PCs based on the test set as well.

The predictive capabilities of each age prediction model were evaluated through the mean absolute error (MAE) metric. MAE provides an immediately comprehensible quantification of the prediction error in terms of how many years off each prediction was on average. The statistical significance of the difference in predictive capabilities across models was assessed through a permutation test. Correlation analyses were performed using the Pearson r correlation coefficient. All statistical analyses and graphical representations were performed using R (https://www.r-project.org/, (accessed on 11 September 2022)).

## 3. Results

### 3.1. Study Selection

Figure 1 shows the full process of the literature search and study selection. In total, 120 reports were initially identified; 55 records were removed before the screening process: 25 were duplicates, 26 articles did not include original research, and 4 articles were not written in English language for a total of 65 remaining records. Of these, 19 records were excluded based on the title and/or abstract (2 studies involved non-human organisms; 17 studies were carried out on tissues different from blood). Twenty-three studies were excluded after a full-text review: 18 studies did not use pyrosequencing technology for methylation analysis, and 5 were carried out on unhealthy people (e.g., growth disorders, Alzheimer’s disease). Of the 23 remaining records, 13 did not provide a publicly available dataset; the study of Garali [16] and Daunay [14] analysed the same dataset, and we retained the dataset of Garali because it was based on an increased number of technical replicates obtained from the same 100 blood samples analysed in Daunay [14]. Consequently, 9 independent datasets from 9 different articles were included in this meta-analysis (Table 2).

The data of the 2298 samples collected in the 9 different datasets included 1147 men and 1151 women whose ages ranged from 0 to 104 years (Appendix A).

### 3.2. PCA Results

Most of the variation in the *ELOVL2* methylation variability was accounted by PC1 (80.3%), which clearly separated studies based on age at the recruitment of the analysed samples (Figure 2A).

PC2 contributed only 11.7% to the total variance and was partially correlated with the pyrosequencing assays used for their detection (Figure 2B). This last observation prompted us to merge all 9 datasets and analyse them jointly for model building and validation purposes, exploiting only CpG sites analysed in both assays (CpG1-CpG7).

### 3.3. ELOVL2 Methylation-Based Prediction Models

With the exception of the values obtained for the dataset of Lucknuch et al. [23], the analysed CpG sites all presented a strong positive correlation with age at recruitment (r > 0.68), indicating that they could all be good estimators of chronological age (Figure 3).

We then tested five different statistical approaches for the development of ELOVL2 single-locus age prediction models (see Materials and Methods). Table 3 reports the classification performances of the tested models. Predicted and chronological ages for each tested model are reported in Figure 4. All tested models showed high prediction accuracy. The model giving the best age prediction accuracy was the gradient boosting regression (GBR) with an MAE of about 5.59 years, while the poorest performing model was multiple linear regression (MLR) with an MAE of about 6.58 years. The support vector machine with radial kernel (SVM) model also showed a high prediction accuracy (MAE = 5.65) with comparable performance with respect to the GBR model (*p* = 0.430). The poor performance of the multiple quadratic regression (MQR) model was mainly due to the influence of outliers with abnormally large errors (Figure 4B).

To evaluate the robustness of the predictive results, a leave-one-out sensitivity analysis was conducted by removing one study at a time (see Appendix A).

Figure 5 reports the sensitivity analysis results for the linear model as a forest plot. We could observe that each time we removed one study, the 95% confidence interval of the MAE overlapped with the 95% confidence interval obtained considering all datasets together. This attests to the robustness of our results. The same thing happened when the other models were considered (Appendix A).

## 4. Discussion

Age estimation using DNA-based methodologies is a crucial step in forensic science analysis, as well as in other fields, such as the monitoring of ageing rate. Although several methods for forensic age estimation have been proposed to date, none of these approaches is currently used in forensic laboratories for identification purposes. In order to translate new discoveries in casework analysis, the definition of precise guidelines for the implementation of the developed methods in a practical manner is a fundamental requirement. To pursue this objective, it is crucial to define a set of methylation markers to be analysed, the relevant methodology for their detection, and an easy-to-use mathematical model for the analysis of the laboratory data allowing for a reliable forensic age estimation. Regarding the definition of the markers, many candidate loci have been proposed, such as ELOVL2, C1orf132, TRIM59, FHL2, KLF14, PDE4C, ELOVL2, FHL2, EDARADD, ASPA, and PENK [8,28,29,30]. As it pertains to the detection methodology, different sequencing/typing techniques have been proposed for forensic age prediction. They include pyrosequencing [15,25,31,32,33], massive parallel sequencing (MPS) [34,35,36,37], SNaPshot assays [38,39], and EpiTYPER [13,28,40,41]. It is important to point out that methylation profiles obtained with these different sequencing/typing methods provide largely comparable results [42]. On the other hand, MPS seems to be the most advantageous approach due to its capability of dealing with low quantity/degraded samples, which can be very common in forensic investigations [43]. Furthermore, MPS is already used in most forensic laboratories for DNA profiles with STR markers, but also for biogeographical ancestry information, mitochondrial DNA sequence analysis, and for forensic DNA phenotyping applications [44,45,46]. Regarding the different algorithms that have been formulated, the Machine Learning approach significantly outperforms other approaches [8,24,34,47].

Systematic reviews carried out in the last years identified hundreds of age prediction models based on DNA methylation data [1,48,49]. These models relied on different tissues (blood or other body fluids) and included fewer than a dozen markers using mainly pyrosequencing to several tens or hundreds of loci using methylation arrays [4,5,6]. A variety of different epigenetic models exploiting different DNA methylation technologies and different statistical methods for forensic age prediction have been developed to date [8,14,15,20,24,50,51,52]. Among them, the most accurate provide prediction errors of 3–4 years, which are in line with those from eyewitness reports. Most of them are based on multiple CpG sites from blood samples for which donors were restricted to adult age ranges, while only a few models covered a full spectrum of human ages from childhood to old age [24,47]. Only few attempts to simplify such epigenetic models have been proposed to date to make them easily applicable in forensic casework [16,53]. These were mainly based on (i) a reduction in the number of markers and (ii) a technological format suitable for forensic laboratories [38,39] resulting in a simple statistical approach (e.g., liner regression) applicable to the data collected in routine practice. Among these several attempts, those recently proposed by Garali and co-workers seem to fulfil all previously mentioned conditions [16]. The proposed single-locus model was based on the seven CpGs sites of the ELOVL2 promoter and showed a prediction error of about 5 years. In addition, despite the fact that multi-locus age prediction models seem to generally perform better than the proposed single-locus model, in independent validation studies, this difference became negligible [54].

In the present meta-analysis, we included nine studies involving more than 2200 participants from different populations to build a single-locus ELOVL2-based epigenetic model of forensic age prediction from blood samples. This allowed us to obtain the largest dataset ever analysed, as well as to improve the understanding of the impact of epigenetic variability of ELOVL2 on forensic age prediction. By using five different statistical approaches, we then compared the differences in the performances obtained using the five different corresponding models. The models giving the best age prediction accuracies were the GBR and the SVM models with a prediction error of about 5.6 years. Sensitivity analysis showed that this error remained stable, indicating that the results obtained were robust.

The *ELOLV2* single locus model was also proposed by two previous studies. The first study was carried out by Garali et al. [16] and was based on a smaller sample size (1413 individuals) with a different methodology. The second study was reported by Zbieć-Piekarska et al. [53] who developed an epigenetic model based on the pyrosequencing of the promoter region of ELOVL2 from 303 blood samples. However, our meta-analysis provides more robust and clearer results since it included new additional studies involving more than 2000 participants. With respect to the study carried out by Garali et al. [16], the classification performances reported in the present study are slightly lower, and this discrepancy might be partially due to overfitting. Garali et al. [16] used every combination of the seven CpGs sites during model building, developing a total of 17,018 age prediction models. This procedure might have overfitted the data, finally resulting in poorer performance when the models are applied in independent validation studies.

For comparative purposes, we performed an additional analysis. First, we applied the best performing models developed by Garali et al. [16] to each study (Table 4). Then, we derived a single best model for each training set obtained by holding out one study at a time from the entire sample set, and we assessed these models based on their respective holdout sets (see Appendix A). Table 4 reports the resulting MAE values. Notably, the models developed by Garali et al. [16] performed well with the datasets that were included in their original study, while they performed much worse with unseen data.

Another important point to consider for the formulation of a single-locus age prediction model is tissue specificity. In fact, even if ELOVL2 methylation levels did not show tissue specificity [18], a significant performance reduction was evident when the obtained models were applied to tissue different from blood. Using methylation data from buccal swab samples of Becker et al. for German and Japanese populations [50] and exploiting the five previously reported statistical models, we obtained prediction errors in terms of the MAE ranging from 17.84 years for the GBR model to 22.7 years for the MLR model for the German samples; similar results were also obtained for the Japanese samples (see Appendix A). These results support the idea that age-prediction models may be cohort- and tissue-specific, and thus, caution should be exercised during their application. Population-specific differences in DNA methylation patterns and their impact on forensic age estimations have already emerged from several published studies on this topic [22,55]. However, the prediction error obtained using the ELOVL2-based epigenetic models is not sufficiently low to make them suitable for forensic practice. This suggests that alongside ELOVL2, the inclusion of additional non-redundant markers is a fundamental requirement to apply molecular models to forensic application with robust results. For instance, starting from the observation that the prediction accuracy of an epigenetic clock is influenced by the proportions of naive and activated immune blood cells [22,56], in a recent study, it was demonstrated that a molecular clock based on ELOVL2 together with a biomarker of immunosenescence (sjTREC) showed a significantly improved prediction accuracy, especially at old ages [22,57] where most epigenetic clocks may become less accurate [15,34,35,58,59].

On the other hand, the formulation of a single-locus epigenetic model represents an easy and cost-effective approach since methylation levels of such candidate regions can be assessed using PCR methods [17,60,61]. In fact, methylation analysis is usually carried out using different DNA methylation technologies (e.g., EpiTYPER^®^, SNaPshot, or pyrosequencing) for which high costs represent a constraining factor for most forensic laboratories. As a cost-effective approach, this last strategy might allow a re-analysis of the same blood sample, a procedure that has been demonstrated to clearly improve the detection of the methylation status of the analysed CpG sites and consequently the corresponding age prediction models [16]. Further studies are required, as also highlighted by Garali et al. [16], to validate the single-locus model we proposed based on DNA samples from different types of tissues to define the applicability of these models to such samples. It might also be interesting to study the capability of this marker to gauge the individual rate of ageing and to evaluate the effects of specific interventions [62,63,64,65].

## 5. Conclusions

In conclusion, in line with a consistent series of epigenetic studies, the findings reported here strongly support the use of *ELOVL2* for the formulation of molecular models of forensic age prediction. Using several machine learning algorithms, in the present studies, we demonstrated that ELOVL2, based epigenetic clocks, shows high prediction accuracy with a prediction error of about 5.5 years with the best performing model. Based on blood samples covering a full spectrum of human ages, the proposed models are thus more suitable for forensic applications.

## Figures and Tables

**Figure 1 ijms-24-02254-f001:**
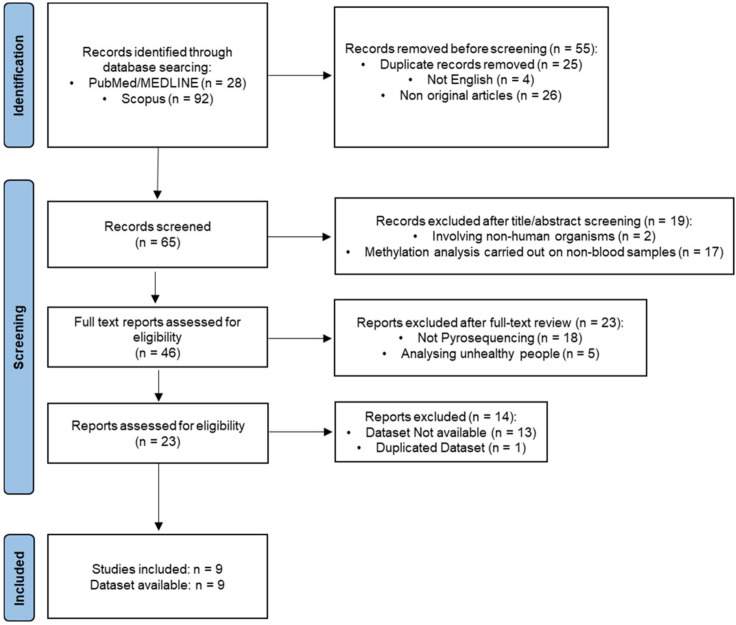
Literature search and study selection process for the systematic review and meta-analysis.

**Figure 2 ijms-24-02254-f002:**
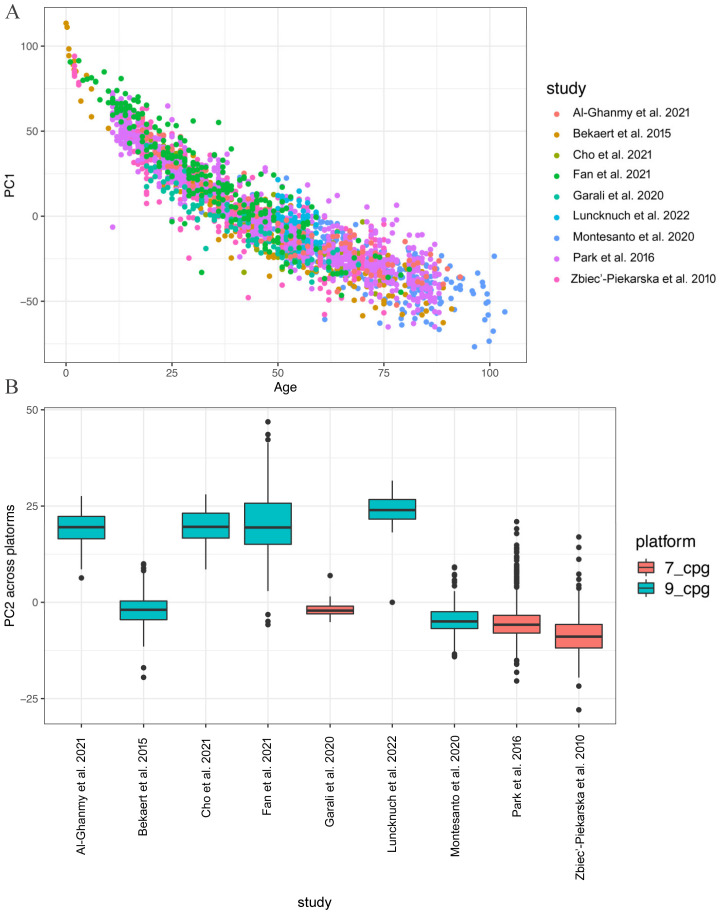
(**A**) Plot of PC1 versus age at recruitment, coloured by study; (**B**) plot of PC2 versus the assay. Source: [8,15,16,20,21,22,23,24,25].

**Figure 3 ijms-24-02254-f003:**
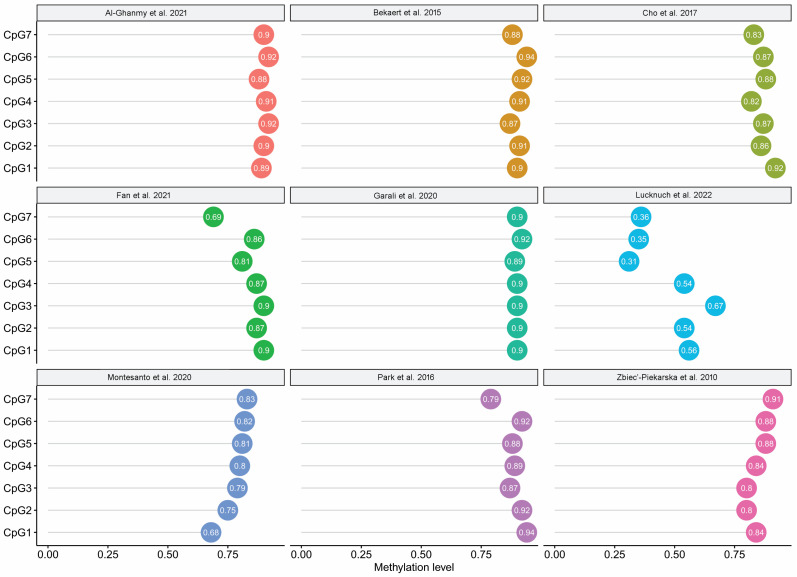
Correlation between chronological age and DNA methylation for the seven/nine CpGs analysed, located in the ELOVL2 promoter in the nine analysed datasets. Source: [8,15,16,20,21,22,23,24,25].

**Figure 4 ijms-24-02254-f004:**
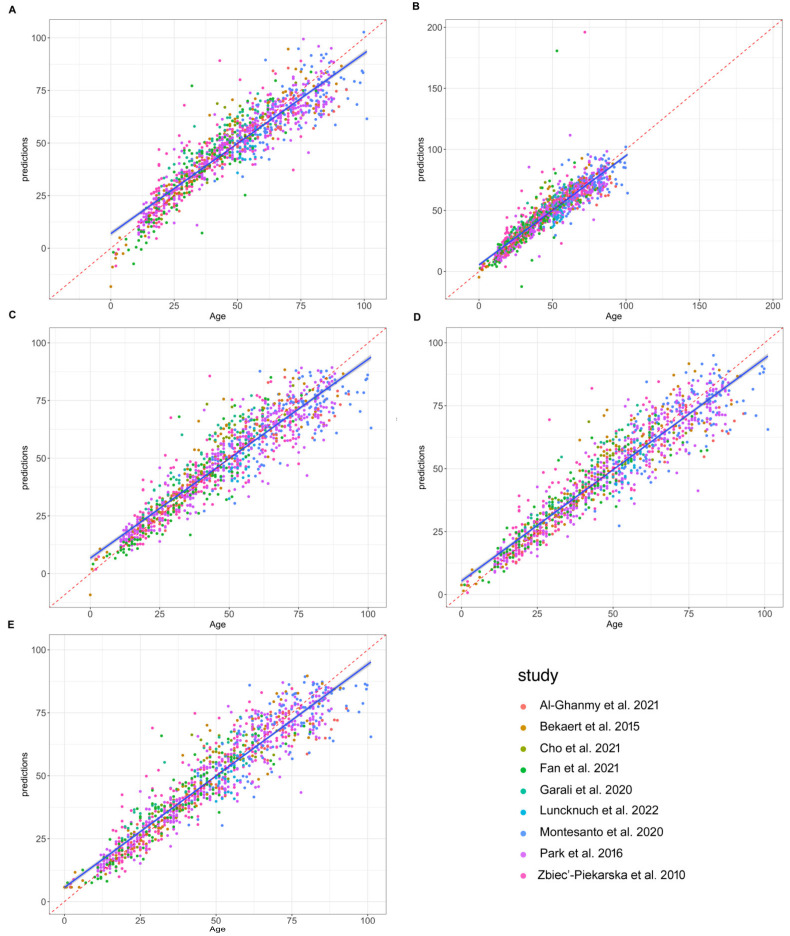
Scatterplots contrasting observed and predicted ages corresponding to five statistical approaches: Multiple Linear Regression (**A**); Multiple Quadratic Regression (**B**); Principal Components (**C**); Support Vector Machine with radial kernel (**D**); Gradient Boosting Regression (**E**). Each dot corresponds to a single prediction. Source: [8,15,16,20,21,22,23,24,25].

**Figure 5 ijms-24-02254-f005:**
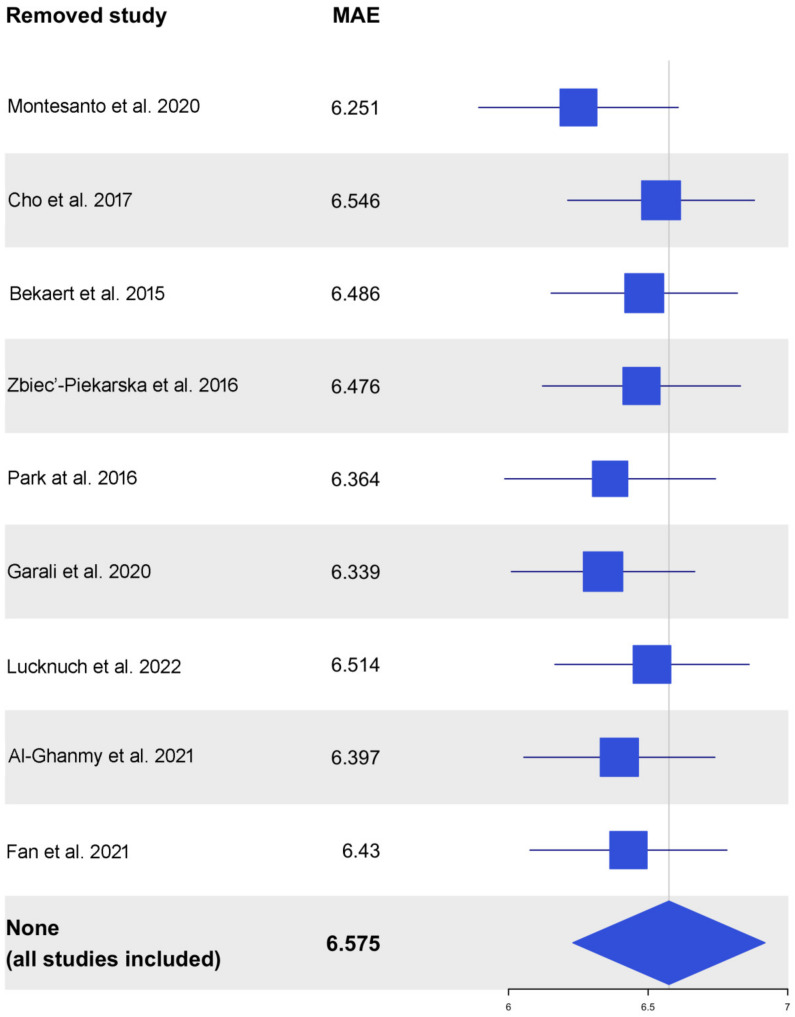
Forest plot presenting the results of the sensitivity analysis for the linear model. Each row reports the results obtained removing a single dataset, except for the final row, which presents the results based on the entire sample set. Results are reported as MAE, along with a graphical representation of the MAE and its confidence interval. Source: [8,15,16,20,21,22,23,24,25].

**Table 1 ijms-24-02254-t001:** Words used in literature search.

PubMed/MEDLINE	Scopus
((((ELOVL fatty acid elongase 2[All Fields]) OR ELOVL2[All Fields]))) AND (((Age[Title/Abstract]) OR aging[Title/Abstract]) OR ageing[Title/Abstract]) AND (pyrosequencing[All Fields])	((ALL (ELOVL AND fatty AND acid AND elongase AND 2) OR ALL (ELOVL2))) AND ((TITLE-ABS-KEY (age) OR TITLE-ABS-KEY (aging) OR TITLE-ABS-KEY (ageing))) AND (ALL (pyrosequencing))

**Table 2 ijms-24-02254-t002:** Characteristics of the studies included in the meta-analysis.

Study	Year of Publication	Population	Age Range (Years)	Sample Size	Reference
Al-Ghanmy et al.	2021	Iraqi	18–93	92	[20]
Bekaert et al.	2015	Belgium	0–91	206	[21]
Cho et al.	2017	Korean	20–74	100	[22]
Fan et al.	2021	Chinese	1–81	240	[8]
Garali et al.	2020	French	19–65	100	[16]
Lucknuch et al.	2022	Thailand	5–60	52 *	[23]
Montesanto et al.	2020	Italian	20–89	323 **	[24]
Park et al.	2016	Korean	1–100	765	[25]
Zbieć-Piekarska et al.	2015	Polish	2–75	420	[15]

* Only control subjects; ** excluding missing values for ELOVL2 methylation data.

**Table 3 ijms-24-02254-t003:** Classification performances of the tested models. The elements on the principal diagonal represent the MAE values of the corresponding model. The content in the upper triangle is represented by the p-values from the t-test comparing the performances of the two corresponding models. The content in the lower triangle is represented by the difference (95% CI in parenthesis) between the performances of the two compared models.

	MLR	MQR	SVM	GBR	PC
**MLR**	6.575	0.1174	<0.001	<0.001	0.0118
**MQR**	0.256 (−0.083, 0.558)	6.319	<0.001	<0.001	0.8826
**SVM**	0.93 (0.708, 1.151)	0.674 (0.351, 1.042)	5.645	0.4295	<0.001
**GBR**	0.989 (0.766, 1.213)	0.733 (0.385, 1.131)	0.059 (−0.087, 0.206)	5.586	<0.001
**PC**	0.226 (0.048, 0.404)	−0.029 (−0.376, 0.367)	−0.703 (−0.908, −0.492)	−0.762 (−0.971, −0.554)	6.348

MLR: multiple linear regression; MQR: multiple quadratic regression; PC: principal components; SVM: support vector machine with radial kernel; GBR: gradient boosting regression.

**Table 4 ijms-24-02254-t004:** Comparison between the best model derived by Garali et al. [16] and the best models we derived by holding out one study at the time. For each holdout study, we report the best model that we identified based on the remaining data, the MAE obtained based on the holdout, and the MAE obtained based on the same holdout based on the best models identified in the Garali study.

Holdout Study	Best Model	Best MAE	MAE Garali SVM 6,7	MAE Garali GBM 6,7 *
Montesanto et al. [24]	SVM	7.77	7.380	7.976
Cho et al. [22]	GBR	5.214	13.666	13.293
Bekaert et al. [21]	SVM	5.984	4.478	4.355
Zbieć-Piekarska et al. [15]	SVM	6.136	5.399	4.398
Park et al. [25]	SVM	7.303	5.871	4.924
Garali et al. [16]	GBR	8.572	4.173	5.430
Lucknuch et al. [23]	GBR	7.892	27.301	27.865
Al-Ghanmy et al. [20]	SVM	9.218	24.452	23.468
Fan et al. [8]	SVM	7.367	13.125	15.275

* Due to the random nature of the Gradient Boosting procedure, it was impossible to replicate the exact same model as in the original publication.

## Data Availability

All data used in this study are provided as Appendix A. The code for replicating the main results is also provided, along with the developed predictive models for application to future studies.

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
