# Peer review of "An ELOVL2-Based Epigenetic Clock for Forensic Age Prediction: A Systematic Review"

_ijms, 2023, doi:10.3390/ijms24032254_

Round 1
Reviewer 1 Report
The work is well developed, and the experimental design seems appropriate.
Author Response
We would thank the Reviewer for appreciating our work.
Reviewer 2 Report
Comments to the author
Manuscript Number ID IJMS-2127076
An ELOVL2 based epigenetic clock for forensic age prediction: a systematic review
This review has described the prediction of chronological age from methylation-based biomarkers represents one of the most promising applications in the field of forensic sciences. The age prediction models so far developed were not easily applicable in forensic casework and the findings here reported strongly support the use of ELOVL2 for the formulation of single-locus epigenetic models, but the inclusion of additional, non-redundant markers is a fundamental requirement to apply a molecular model for forensic application. Although this information is important for studying.
I suggest the quality of this manuscript is good for publication in the International Journal of Molecular science. but it needs some minor grammatical corrections are required before publication. The following are comments for the authors.
In Abstract
Line 21, ‘the’ should be removed
Line 23, ‘caseworks’ to be change ‘caseworkers’ and ‘Pursuit’ are replaced with ‘Pursue’
Line 29 removed ‘that’
Line 31 ‘he gradient boosting’ change to ‘the gradient boosting’
Line 49 Providing change to Provide
Line 51 Removed ‘the’
Line 53 “chronological age several authors also taken advantage” has to change “chronological age several authors have also taken advantage”
Line 71 Remove “Part”
In Result section
Line 86 ‘studies carried out’ change ‘studies were carried out’
Line 103 ‘on the basis of’ should be change ‘based on’
Line 108 Remove ‘for’
149 – ‘expect’ it may change “except”
In Discussion section
Line 153 Predication change to Prediction
Line 157 delete ‘in order’
Line 163 182 and 200- Delete ‘In fact’
In Material and method section
Line 231 abstract change to ‘abstracts’
Line 236 modelling correcte as ‘modeling’
Line 236 ‘on the basis ‘Correcte as ‘based on’
Line 240 ‘a number of’ change as ‘Several’

Author Response
We thank the referee for all these suggestions.
Reviewer 3 Report
The authors presented an interesting study that used DNA levels of the ELOVL2 gene in blood samples to develop an accurate single-locus model for age prediction. The authors developed a systematic review of different published pyrosequencing datasets focused on the ELOVL2 gene, including nine papers from the 65 records.
This manuscript is clearly written, and consistent with the previous studies in this field. The review was written in good English. Figures and tables are well introduced in the manuscript, but the authors should improve the quality of their figures, and add some details to their captions. Please see the comments below.
There is a potential benefit to the publication of this study. However, before publishing, the authors may consider some revisions:
-2. Results (line 79, page 2): If possible, improve the quality of your image on Figure 2 (line 106, page 4), Figure 3 (line 117, page 4), and Figure 4 (line 135, page 5). In addition, the legend in Figure 2 and Figure 4 is also of very bad quality.
-2. Results (line 106, page 4): In the legend of Figure 2, please add, for in each study, the “et al.” and the corresponding reference number. For instance, see Cho et al. [21]. The authors should do the same in Figure 3, Figure 4, Figure 5, and Table 3 (for each study, add the “et al.” and the corresponding reference number to each study).
-2. Results (line 125, page 4): “…while the poor performing model was the PC with a MAE of about 6.58 years.” Please confirm this result. The poorest performing model was the MLR according to table 2. Isn´t it?
-2. Results (line 134, page 5): Remove the sentence on line 134: “Predicted and chronological ages for each tested model were reported in Figure 4.” and put this sentence on line 122 (page 4), after the sentence “Table 2 reports the classification performances of the tested models.”
-2. Results (line 123, page 4): When you refer to MLR, MQR, SVM, GBR on the sentence “All tested models showed high prediction accuracies. The model giving the best age prediction accuracy was the GBR with a MAE of about 5.59 years…. poor performance of the MQR model was mainly due to the influence of outliers with abnormally large errors (Figure 4B).”, you should put the meaning of each acronym. In addition, substitute “accuracies” by “accuracy” in the sentence ““All tested models showed high prediction accuracies.”
-2. Results (line 129, page 5): You should add a legend on Table 2 with the meaning of MLR, MQR, SVM, GBR…
-3. Discussion (line 182, line 187, line 191, page 7): Please add the reference number after the citation of “Garali et al”. It should appear as “Garali et al. [16]”.
-3. Discussion: Please include more studies to allow for a more accurate comparison of the results and to improve your manuscript.
-4. Material and Methods (line 210, page 7): Please put this section “4. Material and Methods” after the “5. Conclusions” section, or in the scope of the manuscript, after the “1. Introduction” section.
-5. Conclusions (line 271, page 9): Please improve our conclusion section.
-Supplementary material: Please remove the cited reference from the supplementary material. This means, the reference “1. Garali, I., et al., Improvements and inter-laboratory implementation and optimization of blood-based single-locus age prediction models using DNA methylation of the ELOVL2 promoter. Sci Rep, 2020. 10(1): p. 15652.” cited at the supplementary material should be removed from the supplementary file. However, the reference citation made in this file “…was identified by applying these models on the validation set. The optimal models for comparison against Garali et al. [1] were…” should be maintained and the corresponding number of the reference included on “[]” should be the same of the corresponding manuscript, in accordance with the list of references at the manuscript file. This means, Garali et al. should be refereed to as [16] in supplementary file, according to the reference number in the manuscript.
Author Response
The authors presented an interesting study that used DNA levels of the ELOVL2 gene in blood samples to develop an accurate single-locus model for age prediction. The authors developed a systematic review of different published pyrosequencing datasets focused on the ELOVL2 gene, including nine papers from the 65 records.
This manuscript is clearly written, and consistent with the previous studies in this field. The review was written in good English. Figures and tables are well introduced in the manuscript, but the authors should improve the quality of their figures, and add some details to their captions. Please see the comments below.
There is a potential benefit to the publication of this study. However, before publishing, the authors may consider some revisions:
-2. Results (line 79, page 2): If possible, improve the quality of your image on Figure 2 (line 106, page 4), Figure 3 (line 117, page 4), and Figure 4 (line 135, page 5). In addition, the legend in Figure 2 and Figure 4 is also of very bad quality.
-2. Results (line 106, page 4): In the legend of Figure 2, please add, for in each study, the “et al.” and the corresponding reference number. For instance, see Cho et al. [21]. The authors should do the same in Figure 3, Figure 4, Figure 5, and Table 3 (for each study, add the “et al.” and the corresponding reference number to each study).
As suggested by the reviewer, in the revised version of our Ms. the quality of all images has been improved and the relevant legends have been adjusted in accordance to his/her suggestions.
-2. Results (line 125, page 4): “…while the poor performing model was the PC with a MAE of about 6.58 years.” Please confirm this result. The poorest performing model was the MLR according to table 2. Isn´t it?
The referee is correct. We apologize for this inconvenience.
-2. Results (line 134, page 5): Remove the sentence on line 134: “Predicted and chronological ages for each tested model were reported in Figure 4.” and put this sentence on line 122 (page 4), after the sentence “Table 2 reports the classification performances of the tested models.”
We thank the reviewer for this observation.
-2. Results (line 123, page 4): When you refer to MLR, MQR, SVM, GBR on the sentence “All tested models showed high prediction accuracies. The model giving the best age prediction accuracy was the GBR with a MAE of about 5.59 years…. poor performance of the MQR model was mainly due to the influence of outliers with abnormally large errors (Figure 4B).”, you should put the meaning of each acronym. In addition, substitute “accuracies” by “accuracy” in the sentence ““All tested models showed high prediction accuracies.”
We thank the referee for all these suggestions
-2. Results (line 129, page 5): You should add a legend on Table 2 with the meaning of MLR, MQR, SVM, GBR…
Table 2 now includes all this information
-3. Discussion (line 182, line 187, line 191, page 7): Please add the reference number after the citation of “Garali et al”. It should appear as “Garali et al. [16]”.
Again, we thank the reviewer for this observation.
-3. Discussion: Please include more studies to allow for a more accurate comparison of the results and to improve your manuscript.
In the revised version of our Ms. the discussion section now includes further details that allow for a more accurate comparison with the results obtained by using other epigenetic models for forensic age prediction.
-4. Material and Methods (line 210, page 7): Please put this section “4. Material and Methods” after the “5. Conclusions” section, or in the scope of the manuscript, after the “1. Introduction” section.
In the revised version of our Ms. section “4. Material and Methods” now is after the section “5. Conclusions”.
-5. Conclusions (line 271, page 9): Please improve our conclusion section.
In the revised version of our Ms. the conclusion section was improved by adding by adding further details regarding the main findings of our work and relevant references.
-Supplementary material: Please remove the cited reference from the supplementary material. This means, the reference “1. Garali, I., et al., Improvements and inter-laboratory implementation and optimization of blood-based single-locus age prediction models using DNA methylation of the ELOVL2 promoter. Sci Rep, 2020. 10(1): p. 15652.” cited at the supplementary material should be removed from the supplementary file. However, the reference citation made in this file “…was identified by applying these models on the validation set. The optimal models for comparison against Garali et al. [1] were…” should be maintained and the corresponding number of the reference included on “[]” should be the same of the corresponding manuscript, in accordance with the list of references at the manuscript file. This means, Garali et al. should be refereed to as [16] in supplementary file, according to the reference number in the manuscript.
Again, we thank the reviewer for this observation.

Round 2
Reviewer 2 Report
Comments to the author
Manuscript Number ID IJMS-2127076
An ELOVL2 based epigenetic clock for forensic age prediction: a systematic review
This review has described the prediction of chronological age from methylation-based biomarkers represents one of the most promising applications in the field of forensic sciences. The age prediction models so far developed were not easily applicable in forensic casework and the findings here reported strongly support the use of ELOVL2 for the formulation of single-locus epigenetic models, but the inclusion of additional, non-redundant markers is a fundamental requirement to apply a molecular model for forensic application.
The manuscript has been improved, but the author needs to re-check some minor corrections in the contain section.
Author Response
We carefully re-checked the minor corrections suggested by the reviewer. We apologize for this inconvenience.